# Exploiting a natural instance of vertebrate-posed chemical aposematism for tick bite prevention: Repellency of *Ixodes scapularis* with (*E*)-Oct-2-enal

Eric L. Siegel[1], Sophia Goodnow[1], Lucy Thompson[1], Sarah Nicolson[1], Elizabeth MacLeod[1], Andrew Y. Li[2], Guang Xu[1], Stephen M. Rich[1]*

**1** Laboratory of Medical Zoology, Department of Microbiology, University of Massachusetts, Amherst, Massachusetts, United States of America, **2** United States Department of Agriculture, Agricultural Research Service, Invasive Insect Biocontrol and Behavior Laboratory, Beltsville, Maryland, United States of America

☯ These authors contributed equally to this work.

* smrich@umass.edu

## Abstract

Ticks are medically important, nuisance arthropods found worldwide. Applications of semiochemical-based tick repellents for personal protection and reproductive/reservoir host-targeted tick interventions are understudied. We evaluated the repellency of a semiochemical allomone of donkey sebum, (*E*)-oct-2-enal, against adult *Ixodes scapularis* ticks – the most frequent human-biting tick in the United States. Ticks were exposed to 20% solutions of (*E*)-oct-2-enal or DEET. A filter paper bioassay was applied under laboratory conditions. Behaviour was observed for 10-min and captured with a tracking software. Changes in velocity and peregrination were assessed relative to negative (ethanol) control groups using multivariable robust regression models. Repellent longevity was defined by preventing ticks from crossing the treated surface and was evaluated as time-to-event data with a Cox proportional hazard regression model. Significant reductions in velocity, increases in peregrination, and strong longevity were observed for both repellents. Overall assessment of repellency indexes with a Principal Component Analysis showed that DEET and (*E*)-oct-2-enal were more effective against male ticks than female ticks. There was no difference in effect on females for each repellent. However, the repellency index for male ticks exposed to (*E*)-oct-2-enal was significantly greater than DEET. This represents the first report, to our knowledge, of the repellency of a natural, vertebrate-emitted semiochemical against *I. scapularis* ticks. Work is needed to understand the underlying mechanism of action of semiochemical repellents. The development of (*E*)-oct-2-enal formulations for practical use in personal protection or reproductive/reservoir host-targeted tick control products warrants further consideration.

**Data availability statement:** The data for this study is available from doi.org/10.17632/hnw69pt9mh.1

**Funding:** Centers for Disease Control and Prevention (CDC U01CK000661).

**Competing interests:** The authors have declared that no competing interests exist.

## 1. Introduction

Ticks present a serious and growing threat to public health and economics in the United States [1]. Serving as vectors for aetiological agents of many human and animal diseases, such as *Borrelia burgdorferi* (Lyme disease), *Anaplasma phagocytophilum* (anaplasmosis), and *Babesia microti* (babesiosis), ticks are also nuisance arthropods to many species of live-stock and wildlife [2–4]. Conventional tick control strategies utilise integrated management approaches. These strategies typically include chemical treatments [5]. For livestock protection, these include chemical acaricides, such as organophosphates and pyrethroids. The effectiveness and applicability of these chemicals are subject to factors such as variable susceptibility of those tick species, emerging patterns of resistance, regional regulations, proper application, and adherence to recommended withdrawal periods for animal products [6,7]. Repellents are almost exclusively used to prevent bites of humans with sparing applications to clothing or skin. These repellents are subject to limitations like products for agriculture use. Additional considerations pertain to user compliance, agreeability, and adverse health effects [8,9]. Consequently, the demand persists for investigating new active ingredients.

Natural products that kill and repel ticks are of particular interest for public health [8,10]. Libraries of botanical extracts have been screened for efficacious oil derivatives, but few studies have considered the protective potential of molecules excreted by vertebrate species that might serve as hosts within the vector's ecological niche [10–13]. Ticks are nearly blind and rely on the perception of subtle chemical stimuli to survive and find a blood meal [14]. The chemical and physical properties of naturally occurring semiochemicals drive the interactions in the tick ecological niche that dictate host- and mate- finding behaviours [15]. The attractive or non-directional (activity-stimulating) effects of volatile semiochemical emissions from vertebrate hosts are well-defined, including those induced by carbon dioxide, ammonia, and fatty acids [16]. The implementation of attractant semiochemicals, largely pheromones of tick origin, can improve acaricide effectiveness [17]. Examples include "lure and kill" modalities that incorporate sex pheromone attractants coupled with impregnated decoy devices. This method functions by increasing the effective contact time of ticks with acaricide-treated surfaces [17,18].

Despite the ubiquity of available vertebrate hosts for many non-nidicolous ticks, variable degrees of specificity have been demonstrated at the species and life stage levels [19]. Some tick species exhibit mild preferences and others may entirely refuse to feed on some vertebrates [15]. Laboratory studies have shown that specific allomones can drive these specificities and be applied in spatial repellent products [12]. This conclusion is only based on limited assessments performed with y-choice olfactometers and in vivo with more specialised tick species. The allomones benzaldehyde and 2-hexanone are demonstrated to repel *Rhipicephalus sanguineus* sensu lato [12,20] The donkey-specific compound, (*E*)-oct-2-enal has also shown strong repellency against *Amblyomma sculptum* and *R. microplus* [13,21]. Recently, (*E*)-oct-2-enal delivered by slow-release devices was tested under field conditions as a repellent against *Amblyomma* ticks [22].

*Ixodes scapularis* is the most frequent human-biting tick in the eastern United States [23,24]. The host-seeking behaviour of *I. scapularis* is defined by a passive, less selective means that seemingly lacks a reliance on vertebrate-emitted olfactory cues. However, *I. scapularis* demonstrates a visible life stage-dependent host specificity. As adults, *I. scapularis* will exclusively feed on a larger reproductive host, primarily the white-tailed deer (*Odocoileus virginianus*). *Ixodes scapularis* will conversely show an aversion to the white-footed mouse (*Peromyscus leucopus*) or eastern chipmunk (*Tomias striatus*). These small mammals are the preferred hosts for the immature stages of the tick, though they may feed on deer and other larger vertebrates at this time [25]. Because this distinction must be made before the

tick initiates feeding, vertebrate-emitted allomones likely influence this specificity. To date, however, semiochemical-based repellency exploiting chemicals excreted by vertebrate hosts has not been demonstrated with *I. scapularis*. The present study sought to assess the response of *I. scapularis* adults to the naturally occurring semiochemical allomone (*E*)-2-oct-enal. Herein, we demonstrate the first instance, to our knowledge, of repelling *I. scapularis* with this vertebrate-emitted allomone. This work sets the foundational work for more targeted research into semiochemical repellents to control *Ixodes* ticks.

## 2. Materials and methods

### 2.1 Tick sourcing and storage

Pathogen free adult *I. scapularis* ticks (150 male and 150 female) were obtained from the tick rearing facility at the Oklahoma State University, Department of Entomology and Plant Pathology, National Tick Research and Educational Resource. Tick sex was confirmed visually. Ticks were received approximately two months after molting and held in a humidity chamber (27-gallon GreenMade professional storage tote with a water base). Ticks were placed in plastic condiment containers on a platform floating in the water. Temperature was maintained at 23 ˚C for storage and experiments.

### 2.2 Chemical preparation

Technical grade (*E*)-oct-2-enal (94%, Sigma-Aldrich, St. Louis, MO, USA; CAS No. 2548-87-0) was used to formulate a test solution. DEET (*N,N-diethyl-meta-toluamide*, Ben's 100 EPA Reg. No. 56575-7; 98.1% active ingredient) was also used as a positive control. These chemicals were diluted to final concentrations of 20% (v/v) in ethanol (absolute, Sigma-Aldrich; CAS No. 16-74-5), consistent with EPA recommendations for testing repellent products intended for use on human skin [26]. Ethanol (absolute) was used as a solvent-only negative control group. Solutions were made fresh every 24 h as needed then vortexed at max speed for 30 s and stored at 23 ˚C. These solutions were mixed by inverting the tube before each use.

### 2.3 Bioassay and working definition of repellency

A filter paper bioassay was modified from previously performed laboratory repellency studies (Fig 1) [10]. A flat lightbox (Noldus XIRWV-4666) emitting white light served as the base for the experimental setup. Other light sources in the room were eliminated to minimise the potential for the influence of light on tick behaviour. Round, 15 cm filter papers (Whatman No. 1 qualitative, VWR International, Radnor, PA, USA; Cat. 28450-150) were divided into three zones by pencil tracing a printed template. The innermost zone was a 4.5 cm inner circle "drop" zone where ticks were introduced. A surrounding 9.0 cm was drawn, measured from the centre of the filter paper. The "treated zone" comprised the annulus of these two circles and was inoculated with 8.25 µL/cm$^2$ of ethanol, DEET, or (*E*)-oct-2-enal solution using a p100 pipette. Outside of the treated zone was an outer, untreated "target" zone spanning the rest of the filter paper, which served as the activity endpoint.

The filter paper was incubated at room temperature for 10 min before being moved to the test site. Ticks were removed from the humidity chamber and placed by the test site for up to 2 h to equilibrate. The ticks were then introduced to their trials in the centre of the drop zone. Ticks were tested individually (n = 300). In total, 50 ticks of each sex were tested per group for ethanol, DEET, and (*E*)-oct-2-enal. A brief inclusion/exclusion evaluation was conducted by placing ticks briefly in the centre of an untreated filter paper at the site of experimentation. Only ticks that moved beyond the drop zone were included in the test. Based on this criterion, up to 20% of ticks evaluated from the population were excluded on the basis that they did not

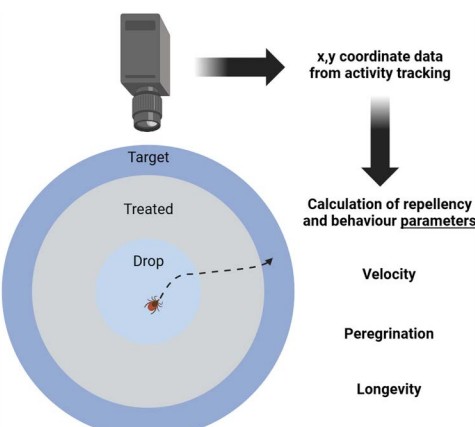

**Fig 1. Diagram of the laboratory filter paper repellency bioassay.** Ticks were placed in the "drop" zone. A concentric "treated" zone was inoculated with 20% (*E*)-oct-2-enal, 20% DEET, or ethanol (absolute). Each tick was considered repelled until it crossed onto the outermost "target" zone. Repellent longevity was defined as the time to reach the target zone.

move immediately when placed on the filter paper. Individual tick behaviour was not tracked during the preliminary evaluation step. Preliminary tests lasted less than 1 min each. Each repellency bioassay trial lasted t = 600 s. Tracking ceased if a tick crossed onto the target zone, and they were considered not repelled. At the end of each trial, ticks were collected and placed in 5 cm petri dishes in the humidity chamber for 24 h. They were briefly screened for mortality by assessing body posture, appendage movement, and response to thermal and respiratory stimuli. Between experiments, the used filter papers were discarded. The lightbox surface was cleaned with RBS 35 concentrate (Cat No. 27950, Thermo Scientific, Waltham, MA, USA) followed by distilled water and wiped clean for each treatment.

## 2.4 Activity tracking

A camera (Basler GenlCam [Basler acA1300-60] (192.168.200.1) was fixed 32 cm above the lightbox and transmitted live video to EthoVision XT, version 15.0 (Noldus Information Technology, Leesburg, VA, USA) for tick behaviour tracking [27]. X,y coordinate data from tick movement was captured at 3.75 frames per sec. These tracks were translated into behaviour parameters for inter-group comparisons, including repellent longevity, velocity, and distance walked (peregrination) parameters. Tick tracks were each reviewed manually and corrected where needed by re-defining erroneously defined frames. Locally estimated scatterplot smoothening (LOESS) was then applied to the x,y coordinates for noise elimination. Merged track visualizations from EthoVision XT are broken down in supplemental information. Velocity was calculated for consecutive x,y frames. The arithmetic mean of velocity measures throughout the trial was computed and assigned to the individual tick as its mean velocity. Without a minimum speed threshold for inclusion, velocity was taken as the integration of time spent moving and true speed while moving.

## 2.5 Statistics and analysis

**2.5.1 Multivariable models for activity parameters.** Raw data from individual ticks were exported from EthoVision XT and analysed using R [28]. The prediction of velocity and peregrination by repellent, tick sex, and the interaction between repellent and tick sex

were assessed with robust regression models [29]. Both models used *M*-estimation in the robust regression framework to minimise the effects of points identified as having large residuals [30]. A log10-transformation was applied to raw velocity and peregrination values to meet model assumptions. An approximation of a pseudo R-squared was calculated using McFadden's method, adjusting for the number of parameters and replacing likelihood with the sum of weighted residuals from the robust models [31].

**2.5.2 Repellent longevity.** Repellent longevity was assessed with a time-to-event analysis using a Cox Proportional Hazard Regression Model [10]. The proportional hazards assumption was checked by plotting the scaled Schoenfeld residuals. This model allowed us to consider with a single parameter both (1) if a tick was repelled; and (2) if not repelled, the time a tick was deterred from crossing the treated zone. The event was defined by the act of crossing into the target zone. Hazard ratios were obtained by exponentiating model coefficients and presented with 95% confidence intervals based on the log-likelihood function. A lower hazard ratio was suggestive of higher repellent efficacy. A hazard ratio of 1 suggested no relative difference between groups. Models were built with an exact approximation to assess the impact of repellent, tick sex, and the interactions between sex and treatment. Confidence intervals for hazard ratios were obtained by using the Wald method, based on the maximum partial-likelihood estimator and Fisher information matrix [32]. Assessing the difference in strength between DEET and (*E*)-oct-2-enal repellency was considered with a confidence interval overlap test of hazard ratios relative to the ethanol baseline. Ticks that were repelled at time t = 600 s were considered right censored. Survival times were depicted in Kaplan-Meier survival curves. A steeper, descending curve was indicative of a weaker repellent effect, while a shallow slope indicated a stronger repellent effect. Median survival times and their non-parametric 95% confidence intervals were calculated using the Brookmeyer and Crowley (1982) method, based on a generalization of the sign test for censored data, using a log-log transformation of the survival function [33].

**2.5.3 Principal component analysis of the three repellency parameters.** The measured parameters (velocity, peregrination, and longevity) were considered together with a principal component analysis (PCA) [34]. A repellency index was calculated based on the first principal component (PC1) score. The contributions of the individual variables to the PC1 were defined by squaring the loadings, dividing by the total variance, and converting to percentages [35]. The correlation matrix was used when conducting the analysis due to the differences in units of measurement for each parameter. Measurement scale differences were addressed by standardizing variables such that mean = 0 and standard deviation = 1. Standard error around PC1 scores was calculated analytically to obtain conservative confidence intervals. Multiple comparisons of mean PC1 scores were made using the Mann-Whitney U test, due to violations of assumptions of normality and homogeneity of variance [36].

## Results

### 3.1 Qualitative observations

Male and female ticks in the ethanol group generally moved straight to the target zone after trial initiation. Some ticks made minor circular movements but did not visually discriminate between the drop and treated zones (Fig 2). Ticks had no trouble navigating the surface treated with ethanol. This was shown by an absence of body posture changes, pauses in peregrination, or sudden directional changes when crossing onto or navigating the treated surface. The activity of ticks exposed to (*E*)-oct-2-enal and DEET, in contrast, was visibly different (Fig 2). These ticks showed immediate aversions to the treatment, even prior to reaching the treatment zone. This was characterised by substantial time probing with elevated front legs

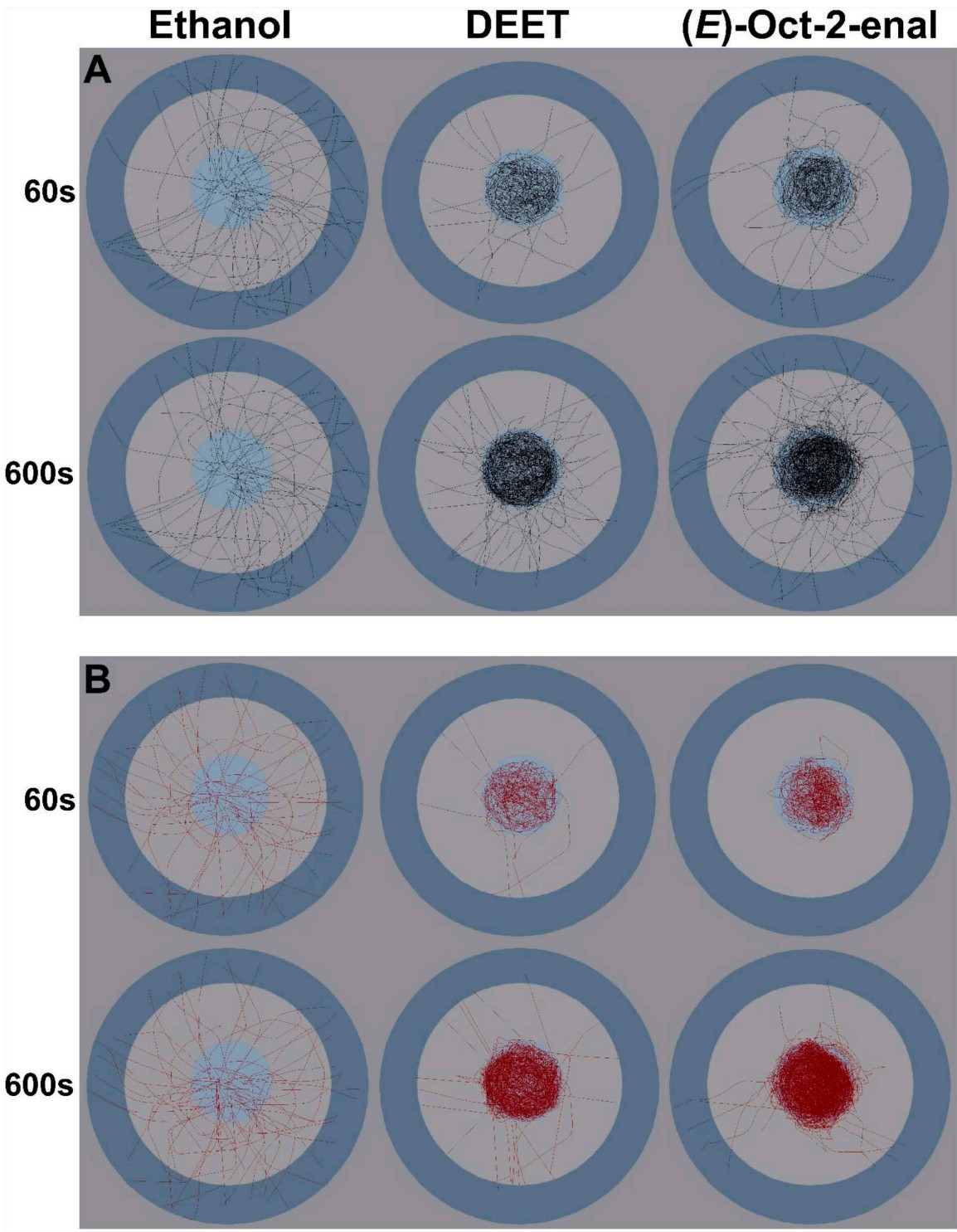

**Fig 2. Longitudinal maps of tick movements.** Tick activity tracks shown at time = 60 s and 600 s for: (A) female *Ixodes scapularis* ticks (black); and (B) male *I. scapularis* ticks (red). Plots represent the collated activity of 50 ticks per group, and each line is the activity of a single tick.

once placed in the drop zone, sudden changes in direction before reaching the treated edge, and retreating from the treated zone upon approach.

Considering the movement on the treated zone in (*E*)-oct-2-enal and DEET trials, no observations were made consistent with irritant effects. It seemed that once ticks made full contact with the treated zone, they navigated across to the target zone without issue. Ticks did not appear to react to DEET until approaching closer to the treated zone when compared to those exposed to (*E*)-oct-2-enal. However, ticks still appeared to avoid physically contacting the DEET-treated surface itself. Ticks exposed to DEET seemed to settle and stop moving in the drop zone after some time if still repelled. Those exposed to (*E*)-oct-2-enal spent more time attempting to overcome the treatment between the drop and treated zones, moving around the drop zone in zig-zag movements as they probed the boundary. No lethality was observed during the trials or through the 24 h post-trial observation period for any groups.

### 3.2 Quantitative changes in behaviour

**3.2.1 Velocity and peregrination.** Tick velocity differed by sex (Fig 3A). Female ticks exposed to ethanol (mean = 0.37 cm/s) moved at velocities faster than males (mean = 0.32 cm/s). A reduction in velocity was seen in ticks exposed to DEET and (*E*)-oct-2-enal due to (1) more time spent still during the trials and (2) slower true movement speed with increased front leg probing activity. Female ticks exposed to ethanol moved an average of 1.9 times faster than those exposed to DEET (mean = 0.19 cm/s). Male control ticks moved 2.7 times faster than DEET-exposed ticks (mean = 0.12 cm/s). Females exposed to ethanol moved an average of 1.7 times faster than (*E*)-oct-2-enal-exposed (mean = 0.22 cm/s). Male controls moved 2.3 times faster than (*E*)-oct-2-enal-exposed (mean = 0.14 cm/s).

Diagnostics of a robust regression model fitted for log-transformed velocity predicted by sex, treatment, and the interaction of sex:treatment indicated acceptable fit: $R^2_{pseudo} > 0.6$; approximately normally distributed residuals (Shapiro-Wilk W = 0.987, p = 0.342); approximate homoscedasticity (Breusch-Pagan BP = 3.24, p = 0.198); no significant multicollinearity with predictors (VIF < 2); and a lack of significantly influential points (Cook's distance < 0.5).

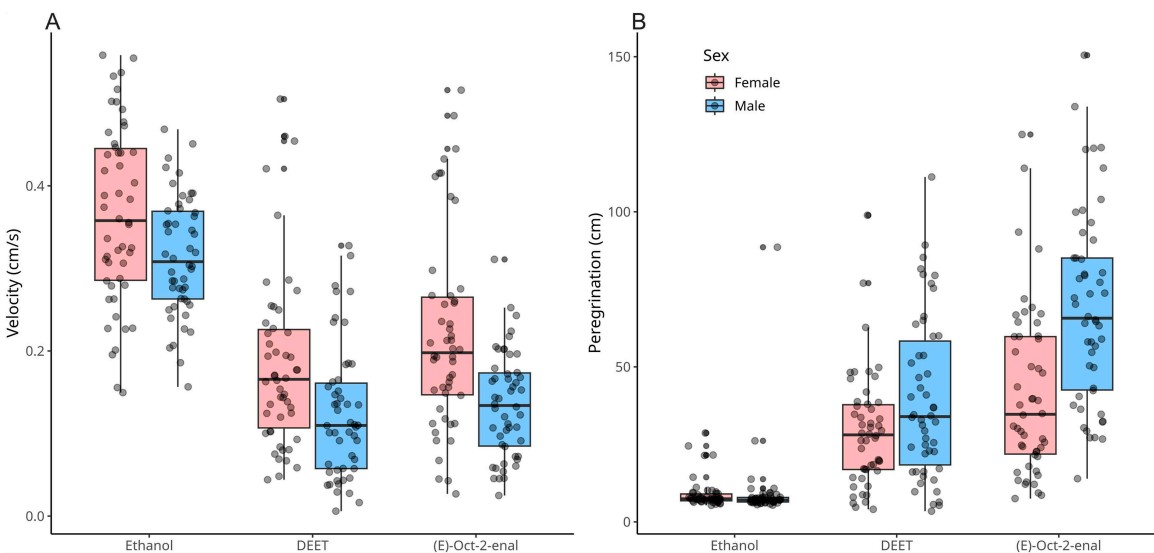

**Fig 3. Quantitative parameters of tick behaviour in repellency bioassays.** Panels note each activity parameter: (A) Velocity; and (B) Peregrination.

This model showed that DEET and (E)-oct-2-enal exposures were significantly predictive of reductions in tick velocity (p < 0.001; Table 1). These reductions were of similar magnitudes for both repellents as shown by marginally overlapping confidence intervals around coefficients (Table 1). Male tick sex was significantly predictive of lower velocity, indicating that male ticks in general had lower velocities than female ticks (p = 0.008). The interaction between male tick sex and DEET or (E)-oct-2-enal each was not significant. This shows that each repellent did not have a stronger reduction in male tick velocity than female (p = 0.330, p = 0.739).

Peregrination of males exposed to ethanol (mean = 9.35 cm) did not differ from that of females (mean = 9.00 cm) (Fig 3B). An increase in peregrination relative to the ethanol baseline was seen with males and females when exposed to (E)-oct-2-enal or DEET. Females exposed to DEET moved 3 times more distance (mean = 29.24 cm) than those exposed to ethanol, and males exposed to DEET covered 4.4 times more distance than those exposed to ethanol (mean = 39.77 cm). This trend was similar with (E)-oct-2-enal. However, the magnitude of the increase was stronger with both males and females when compared to DEET. Females exposed to (E)-oct-2-enal moved 4.6 times more than controls (mean = 41.49 cm). Males moved 7.4 times more distance than ethanol controls, (mean = 68.97 cm).

The regression model fit for the prediction of peregrination had a stronger fit than the velocity model ($R^2_{pseudo}$ > 0.7). Diagnostics, however, similarly showed a normal distribution of residuals (W = 0.992, p = 0.456); homoscedasticity (BP = 2.87; p = 0.238); no multicollinearity (VIF < 1.5 each); and no influential points (Cook's distance max = 0.31). Exposure to (E)-oct-2-enal and DEET was associated with strong increases in peregrination (p < 0.001; Table 1). Male sex itself was not significantly predictive of increased peregrination (p = 0.519), however significant interactions were shown between male sex and (E)-oct-2-enal (p < 0.001) or DEET (p = 0.021) exposures. This indicated that these repellents caused greater increases in peregrination for males than females. The strength of this effect was greater for (E)-oct-2-enal than DEET as noted by a small difference in confidence intervals about respective model coefficients.

**3.2.2 Longevity of repellency (time-to-event).** All ethanol-exposed ticks crossed onto the target zone by the end of the trial. In contrast, 37% of DEET-exposed and 43% of (E)-2-octenal-exposed ticks were repelled at trial conclusion. The majority of these were male ticks: 30 of 50 (60%) DEET-exposed and 37 of 50 (72%) remained repelled, compared to only 7 of 50 (14%) of DEET-exposed and 6 of 50 (12%) of (E)-oct-2-enal-exposed female ticks. Median repellency times exclusive of right-censored ticks for the sum of male and female ticks were much greater (314 s) for DEET-exposed and (E)-2-octenal-exposed (521.5 s) than ethanol-exposed (17.5 s). Repellency times were generally lower for female ticks than male

**Table 1. Multivariable regression models predicting tick velocity and peregrination.**

| Predictor | Velocity | | Peregrination | |
| --- | --- | --- | --- | --- |
| | **Exponentiated Coefficient [95% CI]** | **p (Sig.)** | **Exponentiated Coefficient [95% CI]** | **p (Sig.)** |
| (E)-Oct-2-enal | 0.456 [0.39,0.54] | < 0.001 | 4.420 [3.56,4.48] | < 0.001 |
| DEET | 0.384 [0.34,0.45] | < 0.001 | 3.260 [2.63,3.03] | < 0.001 |
| Sex Male | 0.799 [0.68,0.94] | 0.008 | 0.933 [0.75,1.155] | 0.519 |
| (E)-Oct-2-enal: Sex Male (Interaction) | 0.891 [0.71,1.12] | 0.330 | 1.933 [1.55,2.95] | < 0.001 |
| DEET: Sex Male (Interaction) | 1.041 [0.82,1.13] | 0.739 | 1.413 [1.04,1.19] | 0.021 |

ticks. Female ticks exposed to (E)-2-octenal had a greater median repellency time (217.5 s) compared to those exposed to DEET (151.5 s). The calculation of median repellency times for both DEET and (E)-oct-2-enal-exposed males was not possible as fewer than 50% of ticks crossed onto the target zone by trial conclusion. Repellency times of males that did cross were much greater than females, as observed in Kaplan-Meier survival curves and quantified with right-censored inclusive repellency times (Fig 4, Table 2).

Repellency was further quantified using hazard ratios based on a Cox proportional hazard model (Table 3). Repellency was very strong and comparable for (E)-oct-2-enal (HR relative to ethanol = 0.069) and DEET (HR = 0.071), p < 0.001 each. Male sex was not associated with a significant effect (p = 0.745). The interactions between each repellent and male sex were significant. This indicated that both repellents were generally able to maintain effects longer for male ticks than female ticks ((E)-oct-2-enal P = 0.005; DEET p = 0.027). Though the hazard ratios for (E)-oct-2-enal and sex-stratified (E)-oct-2-enal exposures indicated stronger repellency than DEET, overlap in hazard ratio confidence intervals for each measure suggested that these differences in effects were not significant.

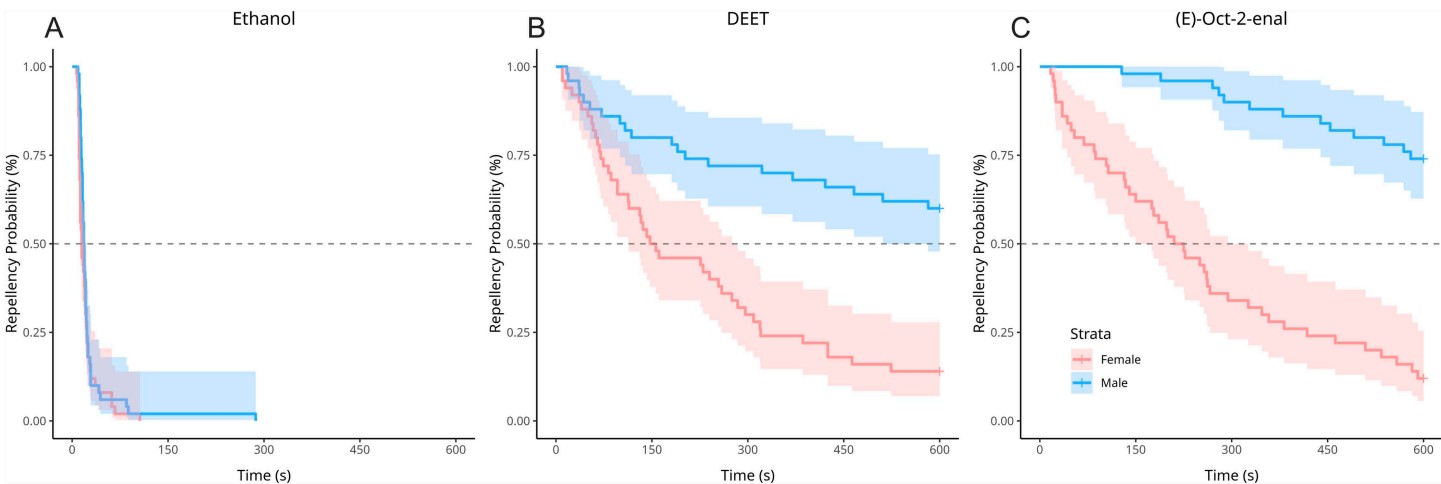

**Fig 4. Kaplan-Meier survival curves indicating the probability of repellency across time.** Repellency was stratified for tick sex across treatments: (A) Ethanol; (B) DEET; and (C) (E)-oct-2-enal. A vertical drop indicates the time a tick crossed the target zone threshold and was considered not repelled. Ticks that were repelled beyond **t** = 600 seconds were considered right censored and repelled. These ticks were incorporated accordingly into the hazard model.

Table 2. Longevity of (E)-oct-2-enal and DEET effects on *Ixodes scapularis* ticks.

| Sex (ticks tested) | Treatment | % ticks repelled at 600s | Median Repellency Time [95% CI] | |
|---|---|---|---|---|
| | | | t = 600 is considered censored | t = 600 is considered an event |
| Male (n = 150) | Ethanol | 0 | 18.5 [16,21] | 18.5 [16,21] |
| | DEET | 60 | Null [Null,Null] | 600 [211,600] |
| | (E)-oct-2-enal | 74 | Null [Null,Null] | 600 [585,600] |
| Female (n = 150) | Ethanol | 0 | 15 [13,20] | 15 [13,20] |
| | DEET | 14 | 151.5 [96,259] | 151.5 [71,320] |
| | (E)-oct-2-enal | 12 | 217.5 [140,266] | 217.5 [91.2,409] |
| All Adult (n = 300) | Ethanol | 0 | 17.5 [15,19] | 17.5 [15,19] |
| | DEET | 37 | 133 [82,183] | 314 [221,406] |
| | (E)-oct-2-enal | 43 | 227 [159,294] | 521.5 [448,595] |

**Table 3. Cox proportional hazard regression model for (*E*)-oct-2-enal and DEET effects on *Ixodes scapularis* ticks relative to the ethanol baseline.**

| Predictor | Hazard Ratio [95% CI] | P (Sig.) |
|---|---|---|
| (*E*)-Oct-2-enal | 0.069 [0.043,0.111] | <0.001 |
| DEET | 0.071 [0.044,0.115] | <0.001 |
| Sex Male | 0.745 [0.501,1.106] | 0.144 |
| (*E*)-Oct-2-enal: Sex Male (Interaction) | 0.448 [0.255,0.787] | 0.005 |
| DEET: Sex Male (Interaction) | 0.531 [0.303,0.931] | 0.027 |

**3.2.3 Repellency index based on a principal component analysis.** A repellency index was calculated for each sex and repellent combination. The PC1 captured 73.45% of variance. The explanation of variance was approximately split among the three factors: peregrination (21.58%), velocity (22.75%), and longevity (29.12%). This indicated that a repellency index based on the PC1 represented a meaningful combination of measured parameters in this study. The repellency indexes were very strong for (*E*)-oct-2-enal and DEET with both tick sexes (Fig 5, Table 4). The standard error in repellent groups, except for male ticks exposed to (*E*)-oct-2-enal, was twice as large as controls, indicating some (but consistently mild) variance in individual-level effects. Repellency indexes for male ticks were significantly stronger than females for both DEET and (*E*)-oct-2-enal (p < 0.001, see Supporting Information). There was no difference in the repellency index for female ticks exposed to DEET and (*E*)-oct-2-enal (W = 1154.5, p = 0.512). However, the difference observed between repellency indexes of male ticks exposed to DEET and (*E*)-oct-2-enal was mildly significant (W = 1002.0, p = 0.041).

## Discussion

Semiochemical-based repellency is rooted in the principle of chemical aposematism, centred around defensive warning signals in nature. Repellent semiochemicals preempt ectoparasite encounters with a given vertebrate species by signaling to the ectoparasite that the vertebrate is unsuitable, or a "nonhost" [4,37]. These emissions contain aromatic/short-chain compounds (e.g., aldehydes or ketones) that do not themselves, however, typically cause biological or reproductive harm to a nearby arthropod. The ectoparasite's ability to distinguish between suitable host and nonhost is particularly important in cases where the geographical ranges of distinct vertebrates overlap. This phenomenon serves as an understudied opportunity to advance arthropod control schemes, in which the signature odour of a vulnerable host can be masked by that of a nonhost. One example of successful application of this principle in other vector systems is the protection of cattle from the tsetse fly (*Glossina* spp.) with controlled release collars emitting mimicking odours of waterbuck (*Kobus defassa*) in East Africa. Another example is protection of chickens from the red poultry mite (*Dermanyssus gallinae*) with the odours of the duck (*Anas platyrhynchos*) uropygial gland [38,39]. Although candidate molecules for such processes have been evaluated against some tick species, there are currently no semiochemical repellent products in use that target ticks [12,13,20,40].

Results on (*E*)-oct-2-enal presented in this report represent the first instance, to our knowledge, of a naturally occurring, vertebrate-emitted semiochemical repelling *Ixodes* ticks. Using a laboratory assay integrated with a behaviour tracking platform, repellent longevity was defined by a time-to-event approach. Behavioural effects were considered based on a combination of activity parameters. The filter paper bioassay was strict on the performance of the repellent treatment, requiring ticks to be confined within a very small starting area.

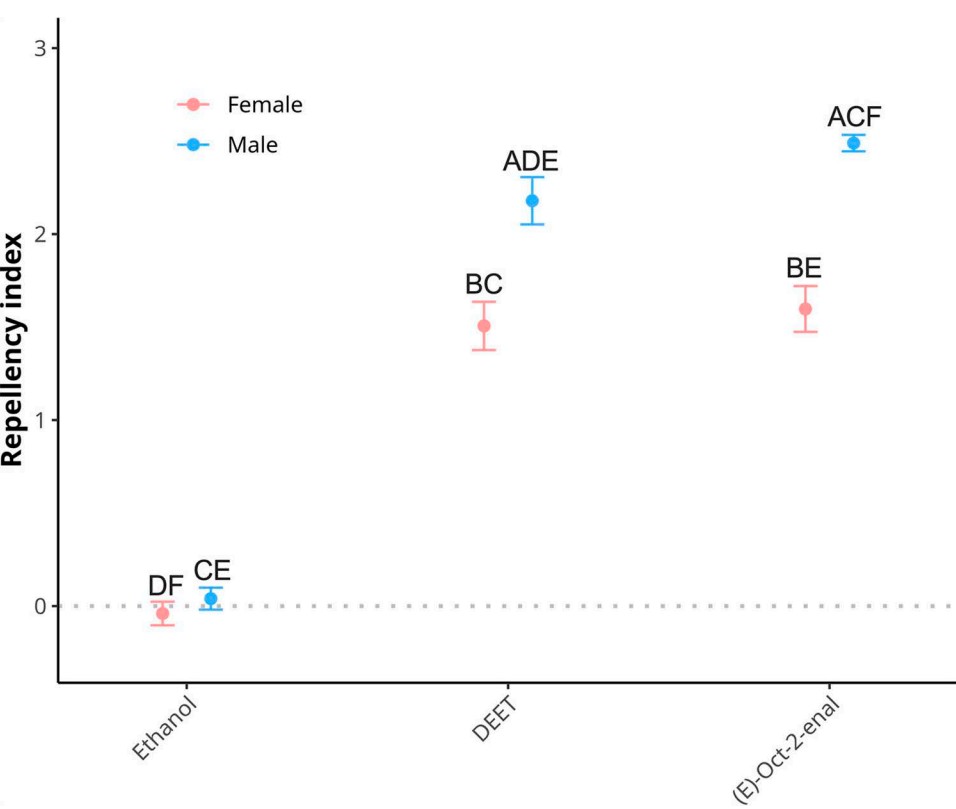

**Fig 5. Visualisation of repellency indices stratified by repellent and tick sex.** Repellency index is based on mean PC1 score. Significance from relevant pairwise comparisons are shown with letters corresponding to groups in Table 4. Comparisons were made between repellents, same sex within repellents, and same sex between repellents. Significance indicates **p** < 0.001, except **p** (C vs E) = 0.041.

Despite this, we observed strong effects on tick behaviour with each parameter. DEET, the gold standard repellent compound for personal protection, was used as a positive control. The effects on tick activity posed by (*E*)-oct-2-enal were similar to those observed with DEET but repellency was stronger with (*E*)-oct-2-enal. Male ticks were more susceptible in general to (*E*)-oct-2-enal than female ticks. Differential (qualitative) behavioural responses to DEET and (*E*)-oct-2-enal were also observed.

Ticks exposed to (*E*)-oct-2-enal appeared to be repelled from greater distances from the treated zone than those exposed to DEET, which is consistent with the differences in vapour pressure between the two molecules that dictate disparate evaporation rates. The vapour

**Table 4. Repellency index calculations based on peregrination, velocity, and longevity.**

| Treatment | Sex | Repellency Index | Standard Error |
|---|---|---|---|
| Ethanol | Male (A) | 0.040 | 0.059 |
| | Female (B) | -0.040 | 0.064 |
| DEET | Male (C) | 2.179 | 0.127 |
| | Female (D) | 1.506 | 0.130 |
| (*E*)-Oct-2-enal | Male (E) | 2.489 | 0.044 |
| | Female (F) | 1.597 | 0.123 |

pressure of (*E*)-oct-2-enal (0.6 mmHg at 20 °C) is about 100x that of DEET (0.0056 mmHg at 20 °C) [41,42]. The tick response to (*E*)-oct-2-enal was characterised by abrupt movement speed changes, aversions, and probing with their front legs prior to reaching the treated surface (as opposed to the elicitation of an irritant effect with effective contact with the treated surface). It may therefore be hypothesised that spatial repellency is responsible for behavioural changes [43]. We are limited to speculation about the repellency mechanism at this point as the bioassay used in this report is not designed to test spatial repellency. It is difficult to visually distinguish these details at this level of observation. DEET for example is a low volatile compound, yet a tick can sense the compound at 1-3 mm from the edge of treated filter paper [44]. Although ticks behave differently to (*E*)-oct-2-enal, we cannot yet make a conclusive determination that this spatial repellency. Nonetheless, the immediacy of the response does suggest that the mode of action bears resemblance to that of semiochemical repellent molecules described against *R. sanguineus* ticks [20].

Over time, ticks became tolerant to the treated zone and crossed without issue. This was especially notable with female ticks. This change in behaviour could be due to the saturation of uncharacterised chemoreceptors of the tick sensory system(s) or the saturation of the still air with the active ingredient around the filter paper. Alternatively, it could be attributed to some level of tolerance development with longer exposures. These hypotheses could be assessed in future work. However, the priority is to determine how much of the repellency is due to contact vs spatial mechanisms by conducting spatial repellency-specific tests (e.g., the static air assay) and sensory electrophysiology experiments. This would guide the application and development studies necessary for formulation assessment for use on humans or animals.

Given the absence of host cues and environmental conditions in this study, we can further speculate that (*E*)-oct-2-enal *at least* functions in the traditional repellent sense, wherein the arthropod demonstrates a directional aversion to the chemical source [45]. We cannot discount other mechanisms at this point that may contribute to protective effects. Some repellents, for example, have been described to interfere with the perception of directional or stimulating host emissions (including $CO_2$ and heat), and others induce intoxication/confusant-type effects against ticks and other arthropods [18]. Conversely, it would be important to know if a tick would overcome the repellent effect with a trumping presence of an appetence-stimulating cue. Stepwise expansion to host cue- and environmental condition-inclusive experiments can help us answer these questions. Other responses may also be elicited with the use of other validated assays, such as a wall climb assay, which would permit detachment consistent with behaviour in nature [8].

Effective chemical protection schemes based on similar patterns of non-host repellent excretions have been detailed in other systems. Examples include the reaction of leopard geckos (*Eublepharis macularius*) and green lizards (*Lacerta viridis*) to the defensive secretions of *Graphosoma lineatum* [46,47]. Prior tick research on repellent semiochemicals report focused on species with relatively high host specificity, including *R. sanguineus, R. microplus,* and *A. sculptum* [25,48,49]. *Rhipicephalus sanguineus* has a high affinity to feed on canids. However, they have been demonstrated to differentially parasitise breeds of dogs (*Canis lupus familiaris*) due to nonhosting features posed by tick-resistant breeds, such as beagles [50]. The decreased susceptibility of beagles to *R. sanguineus* parasitism has been attributed to the beagle's production of two described volatiles, benzaldehyde and 2-hexanone. These volatiles are then recognised by host-seeking ticks as an indicator associated with a strong immune response to tick feeding, which in turn promotes poor biological outcomes (longevity, reproductive success) for the tick [12]. A similar phenomenon has been documented with *A. sculptum,* which will feed on horses (*Equus ferus caballus*) over the donkey (*E. africanus*

*asinus*) in environments where host presence overlaps [49]. This interaction has been shown to be governed by tick recognition of (*E*)-oct-2-enal present in *E. africanus asinus* sebum [13]. Selective breeding of cattle based on phenotypes that resist *R. microplus* infestation with repellent volatile emissions, such as 6-methyl-5-hept-2-one, hexyl acetate, benzaldehyde, and (*E*)-hept-2-enal, is also of interest but still in early stages of research [51].

When compared to the above tick species with high host specificity, *I. scapularis* is much more opportunistic and generally shows a limited host specificity [23]. Given the human-biting affinity of *I. scapularis*, this species is an important target for human personal protection. Ticks have three life stages, larval, nymphal, and adult stages. The nymphal stage of *I. scapularis* is recognised as the primary vector for the causative agents of Lyme disease (*B. burgdorferi* sensu stricto), anaplasmosis (*A. phagocytophilum*), and babesiosis (*B. microti*) due to several reasons that are reflected by the consistency of the seasonal occurrence of disease cases and the seasonal host-seeking activity of the nymphal stage [2]. However, adult, female *I. scapularis* still bite humans and are responsible for smaller peaks in disease agent transmission, shown in the fall months when nymphal ticks are not active [3]. Male ticks are also important in the reproductive maintenance of tick populations and thus important to stable transmission cycles. Aside from their relevance with human-biting activity, adult ticks could also represent a neglected opportunity for off-human application of semiochemical-based repellency. Adult *I. scapularis* will feed on larger hosts, such as *O. virginianus*, capable of providing sites for reproduction [25]. Immature *I. scapularis* in comparison will primarily be supported by small mammals (*P. leucopus*; *T. striatus*), though they may feed on larger species. As adult *I. scapularis* show an aversion to feeding on small mammals, *P. leucopus* and *T. striatus* could be considered nonhosts to *I. scapularis* adults.

Given our initial findings that vertebrate (nonhost) allomones can repel *I. scapularis,* it may be possible to identify an effective repellent semiochemical molecule, or combination of molecules, by considering differences in *O. virginianus* and small mammal volatile profiles through the application of comparative sebomics. It is too soon to tell how effective this approach may be; however, other studies have laid the groundwork for this investigation. These include a study describing volatile compounds from the interdigital gland of male white-tailed deer, which did not include (*E*)-oct-2-enal, and other studies that established that we could induce attractive/arrest behaviour with kairomones extracted from white-tailed deer external glands [52,53]. Volatile emission profiles of small mammal hosts for immature *I. scapularis* have yet to be described. If differences are observed, it may be possible for us to mask the attractive or tolerated odour blend of *O. virginianus* with an unattractive volatile from small mammals and thereby reduce *I. scapularis* burden on their principal reproductive host. This approach could be sustainable and apply natural technologies and materials. The matter of life stage-dependent repellency raises several more questions. As we do not see the same host specificity with immature *Ixodes* ticks as adult ticks, it would also be of interest to determine if the nymphal stage responds to (*E*)-oct-2-enal or other repellent semiochemicals. When this semiochemical was evaluated against *A. sculptum* in a y-choice olfactometer laboratory test, significant repellent activity was verified with nymphs. However, a recent field trial discrepancy showed that this molecule was ineffective against nymphs despite affecting adults [22]. It remains to be seen whether this discrepancy is due to the artefact of the transition to the environmental setting or a true difference in response by life stage.

Many knowledge gaps pertaining to physiology also exist, ranging from the emission of these naturally occurring semiochemicals to repellent perception in the tick. Beginning with vertebrate emission, the reasons for production are not known. For example, (*E*)-oct-2-enal is found in the volatile profile of several vertebrate species, but its purpose

has not been characterised. With regards to perception, Haller's organ is the sensory basis for the tick perception of volatilised host cues [54]. We do not, however, know if Haller's organ-mediated olfaction exclusively accounts for the chemoreception of these molecules or if other structures play major (or minor) roles in perception. The present study also uncovered differences in the strength of response between male and female *I. scapularis* to repellents. Sexual dimorphisms in the gross anatomy of Haller's Organ are found with other species, such as *Dermacentor variabilis*, but have not been shown with the *Ixodes* genus [54]. It would be interesting to see if there are differences in the response at the molecular level between male and female ticks to these compounds. Alternatively, the differences we see could solely be attributed to the differences in size and movement dynamics between male and female *I. scapularis* ticks. We would also benefit from understanding basic questions that remain unanswered at the interface of tick ecology and chemosensory physiology. These include the distances required to elicit a response from the tick, the chemical structures involved in sensing host-emitted repellent semiochemicals, and the influence of pathogen infection (of the host or tick) on emission and response [18]. The matter of how important relative emission profile is in relation to absolute release volumes/concentration has also been previously noted and is also of interest to study further for semiochemical-based formulations [22].

## Acknowledgements

The authors thank Dr. Patrick Pearson for his helpful comments during the preparation of the manuscript. This article reports the results of research only. Any mention of a proprietary product does not constitute an endorsement or a recommendation by the authors or the USDA for its use. The USDA is an equal opportunity provider and employer. The conclusions, findings, and opinions expressed by authors contributing to this journal do not necessarily reflect the official position of the authors' affiliated institutions.

## Author contributions

**Conceptualization:** Eric L Siegel, Guang Xu, Stephen M. Rich.

**Data curation:** Eric L Siegel, Sophia Goodnow, Lucy Thompson, Sarah Nicolson, Elizabeth MacLeod.

**Formal analysis:** Eric L Siegel, Guang Xu.

**Funding acquisition:** Stephen M. Rich.

**Investigation:** Eric L Siegel, Andrew Y Li, Stephen M. Rich.

**Methodology:** Eric L Siegel, Sophia Goodnow, Lucy Thompson, Sarah Nicolson, Elizabeth MacLeod, Guang Xu, Stephen M. Rich.

**Project administration:** Stephen M. Rich.

**Resources:** Andrew Y Li, Stephen M. Rich.

**Software:** Andrew Y Li, Stephen M. Rich.

**Supervision:** Eric L Siegel, Andrew Y Li, Guang Xu, Stephen M. Rich.

**Validation:** Andrew Y Li, Guang Xu, Stephen M. Rich.

**Visualization:** Eric L Siegel, Guang Xu, Stephen M. Rich.

**Writing – original draft:** Eric L Siegel, Guang Xu.

**Writing – review & editing:** Eric L Siegel, Sophia Goodnow, Lucy Thompson, Sarah Nicolson, Elizabeth MacLeod, Andrew Y Li, Guang Xu, Stephen M. Rich.

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
