## [Decision Letter · Decision Letter 0]

7 Nov 2024

PONE-D-24-39388Exploiting a natural instance of vertebrate-posed chemical aposematism for tick bite prevention: Repellency of Ixodes scapularis with (E)-Oct-2-enalPLOS ONE

Dear Dr. Stephen M. Rich,

Thank you for submitting your manuscript to PLOS ONE. After careful consideration, we feel that it has merit but does not fully meet PLOS ONE’s publication criteria as it currently stands. Therefore, we invite you to submit a revised version of the manuscript that addresses the points raised during the review process.

**ACADEMIC EDITOR: **

The study is interesting and the manuscript was well prepared. However, the manuscript needs major revision.

Check and revise “repellency of Ixodes scapularis with (E)-Oct-2-enal” in the title.

Line 356-358: How male and female ticks were identified?. Add details.

Check and write specific subheadings in the results section.

Please submit supplementary files along with the manuscript.

We look forward to receiving your revised manuscript.

Kind regards,

S Ezhil Vendan, Ph.D

Academic Editor

PLOS ONE

Journal Requirements:

“This research and the article publishing charge were funded by the New England Center of Excellence in Vector-borne Disease (CDC U01CK000661).”

“New England Center of Excellence in Vector-borne Disease (CDC U01CK000661)”

Additional Editor Comments :

The study is interesting and the study design and manuscript preparation was well. However, the manuscript needs revision.

Check and revise “repellency of Ixodes scapularis with (E)-Oct-2-enal” in the title.

Line 356-358: How male and female ticks were identified?. Add details.

Check and write specific subheadings in the results section.

Please submit supplementary files along with the manuscript.

Reviewers' comments:

Reviewer's Responses to Questions

**Comments to the Author**

1. Is the manuscript technically sound, and do the data support the conclusions?

Reviewer #1: Yes

Reviewer #2: Yes

2. Has the statistical analysis been performed appropriately and rigorously? 

Reviewer #1: No

Reviewer #2: Yes

3. Have the authors made all data underlying the findings in their manuscript fully available?

Reviewer #1: Yes

Reviewer #2: Yes

4. Is the manuscript presented in an intelligible fashion and written in standard English?

Reviewer #1: Yes

Reviewer #2: Yes

5. Review Comments to the Author

Reviewer #1: I thank the authors for their submission of the manuscript “Exploiting a natural instance of vertebrate-posed chemical aposematism for tick bite prevention: Repellency of Ixodes scapularis with (E)-Oct-2-enal”. As the authors point out, developing new natural tick repellants is important, especially with the emergence of resistance to several classes of repellants and acaricides and the public perception of other compounds. Overall, I find the manuscript to be well written, the experiment to be appropriate to answer the question posed, and the results and interpretations seem reasonable. The statistical analysis used to achieve these results could be more robust.

Major Comments:

1. For peregrination, a linear model that allows multivariate analysis would be more appropriate than several independent t-tests between groups

2. The results for differences in RT50 should include confidence intervals.

3. When reporting the hazard ratios for the DEET groups, the mean and confidence interval estimates do not make sense. The mean hazard ratios are .220 and .2753 for males and females, respectively. However, the mean for all adults is .0618.

4. The Kaplan-Meyer curve should include confidence intervals.

5. The authors mention first evaluating the ticks on untreated filter paper. However, this is excluded from the analysis. This provides an important comparator that allows the authors to account for individual-level variation in tick movement using difference-in-differences analysis. This would correct for any possible accidental bias or random variation in how individuals were assigned to groups.

Minor Comments:

1. In Table 1, Are the right censored exclusive/inclusive switched? It currently reads that median repellency is undefined when they are excluded.

2. Figures 2 and 3 are extremely low quality, to the point that legends are difficult to read.

3. On lines 298-301: There are reasons beyond corresponding phenology that the nymphal tick is suspected to be the life stage most likely to spread B. burgdorferi to humans.

4. On line 335: an “is” is missing.

Reviewer #2: This study examined (E)-oct-2-enal, a natural compound from donkey sebum, as a potential repellent against adult Ixodes scapularis ticks. Compared to DEET, (E)-oct-2-enal showed similar or stronger effects, reducing tick movement speed and increasing exploratory distance in a laboratory bioassay

Line 33: "Cox" should be capitalized.

Line 72: Add an explanation for "aposematism" could aid clarity if unfamiliar to general audiences.

Line 112: "semiochemical" could be clarified along with its type and explain (E)-oct-2-enal falls under which type of semiochemical.

Line 152: Use "times" instead of "x" to improve readability.

Line 292: "(P. leucopus; T. striatus)” Write full names of species when written for the first time in the manuscript.

Line 330: Clarify by specifying "field trial discrepancy."

Quality (pixels) of Figure 1 and Figure 4 should be improved.

6. PLOS authors have the option to publish the peer review history of their article (what does this mean? ). If published, this will include your full peer review and any attached files.

**Do you want your identity to be public for this peer review?** For information about this choice, including consent withdrawal, please see our Privacy Policy .

Reviewer #1: No

Reviewer #2: **Yes: ** Abrar Hussain

---

## [Author Response · Author response to Decision Letter 1]

22 Dec 2024

22 December 2024

The authors thank the academic editor, as well as Abrar Hussain and the other anonymous reviewer, for the valuable comments and care with this manuscript. All queries and concerns have been addressed in full, noted point by point below. The authors are grateful for the improvements made to this manuscript that originate with the below feedback. A marked-up copy of the manuscript with all changes tracked is attached, labeled ‘Revised Manuscript with Tracked Changes.’ These are collated in the file labeled ‘Manuscript.’

Academic editor:

1. 1. Please ensure that your manuscript meets PLOS ONE's style requirements, including those for file naming. The PLOS ONE style templates can be found at (provided links).

The authors acknowledge the provided journal guidelines and have updated the manuscript to conform with these requirements from the original submission formatted for PLoS Biology.

2. We note that you have provided additional information within the Acknowledgements Section that is not currently declared in your Funding Statement. Please note that funding information should not appear in the Acknowledgments section or other areas of your manuscript. We will only publish funding information present in the Funding Statement section of the online submission form. Please remove any funding-related text from the manuscript and let us know how you would like to update your Funding Statement. Currently, your Funding Statement reads as follows: “New England Center of Excellence in Vector-borne Disease (CDC U01CK000661)”

The authors have amended the acknowledgment section to remove funding information. The funding information above is accurate.

The authors note that there may have been an error when completing the data availability statement of the submission form. All data has been made accessible according to the

statement in the supporting information section of the original manuscript, lines 668-677 of the revised version. All data and supplemental files have been placed in the Mendeley database: https://data.mendeley.com/preview/hnw69pt9mh?a=ba5a4312-b421-408b- acd4-3f4e8406cb58. This database has an associated (reserved) DOI: 10.17632/hnw69pt9mh.1. While the database is not published, it is accessible to the reviewers by following the information in the manuscript. Further, this database is updated through the review process. It will be available without restriction to the public if accepted and at that time.

4. The study is interesting and the study design and manuscript preparation was well. However, the manuscript needs revision.

The authors appreciate the academic editor’s feedback. The manuscript has been substantially revised according to all comments. Specifics are acknowledged below.

5. Check and revise “repellency of Ixodes scapularis with (E)-Oct-2-enal” in the title.

The authors have revised the title according to the academic editor’s feedback.

6. Line 356-358: How male and female ticks were identified?. Add details.

The authors have amended section 2.1, lines 106-107 to include information on how ticks were identified: “Ticks were provided with labeled vials, separated by sex. Tick sex was confirmed visually without the need of a microscope.”

7. Check and write specific subheadings in the results section.

The authors acknowledge that specific subheadings in the results section add to organization of the manuscript structure. As such, the results section has been separated by level 2 and 3 subheadings. These correspond to sections outlined within the materials and methods.

8. Please submit supplementary files along with the manuscript.

The authors note that supplementary files may be found at the link in point 3 requested by the academic editor. This Mendeley database is detailed in the supplemental information section at the end of the manuscript.

Reviewer 1:

Feedback: I thank the authors for their submission of the manuscript “Exploiting a natural instance of vertebrate-posed chemical aposematism for tick bite prevention: Repellency of Ixodes scapularis with (E)-Oct-2-enal”. As the authors point out, developing new natural tick

repellants is important, especially with the emergence of resistance to several classes of repellants and acaricides and the public perception of other compounds. Overall, I find the manuscript to be well written, the experiment to be appropriate to answer the question posed, and the results and interpretations seem reasonable. The statistical analysis used to achieve these results could be more robust.

The authors are grateful to the reviewer for the time and care with this manuscript. Queries and recommendations made below have been addressed in full:

Major Comments:

1. For peregrination, a linear model that allows multivariate analysis would be more appropriate than several independent t-tests between groups.

The authors thank the reviewer for the suggestion and agree that the analysis with independent t-tests is not as sound as a regression-based analysis. The authors have changed the analysis of peregrination and velocity to use robust regression models. The results are consistent with the original analysis with regards to the strength/significance of associations. The updated materials and methods, results, and interpretations for these analyses are provided in section 2.5.1, lines 173-181 (method), results section 3.2.1, lines 244-292. To build on this, the parameters were considered together with a principal component analysis to obtain a repellency index based on the first principal component. This can be found in the methods, section 2.5.3, lines 201-211, and in the results, section 3.2.3, lines 332-353, table 4, figure 5.

2. The results for differences in RT50 should include confidence intervals

The authors thank the reviewer for suggesting to improve the survival analysis by including confidence intervals about the 50% repellency time. These have been calculated and are presented in Table 2, line 316-318. Additionally, the Kaplan-meier survival curves have been updated to show the confidence intervals visually with bands around the group trend (Figure 4).

3. When reporting the hazard ratios for the DEET groups, the mean and confidence interval estimates do not make sense. The mean hazard ratios are .220 and .2753 for males and females, respectively. However, the mean for all adults is .0618.

The authors acknowledge that the method used for obtaining hazard ratios for male, female, and adult with 3 separate models affected the consistency of hazard ratios due to the different number of censored ticks. To fix this issue, A single Cox model was performed considering the interaction of sex and repellent to draw inference on the same data with a more sound and straightforward interpretation. This has been updated in table 3, lines 229-231, and in text lines 319-327.

4. The Kaplan-Meyer curve should include confidence intervals.

The authors thank the reviewer for the suggestion to improve figure 4, Kaplan meier survival curves. Figure 4 is now organised into three panels (by repellent instead of by sex) in order to have 2 lines per graph not 3 (clearer when adding confidence interval bands).

5. The authors mention first evaluating the ticks on untreated filter paper. However, this is excluded from the analysis. This provides an important comparator that allows the authors to account for individual-level variation in tick movement using difference-in- differences analysis. This would correct for any possible accidental bias or random variation in how individuals were assigned to groups.

The authors thank the reviewer for raising this point that was not clear in the methods section. The methods section has been updated in section 2.3, lines 142-444 to clarify the limited scope of the inclusion/exclusion criterion that did not permit collection of data. tick selection was made at random for group allocation.

Minor comments:

1. In Table 1, Are the right censored exclusive/inclusive switched? It currently reads that median repellency is undefined when they are excluded.

2. The authors have revised the table (now table 2) to correctly differentiate (and label more clearly) the result of ticks with repellency at t = 600 sec. Additionally, confidence intervals have been added. Errors presenting mean instead of median for censored inclusive estimates have also been revised.

3. Figures 2 and 3 are extremely low quality, to the point that legends are difficult to read.

The authors thank the reviewer for pointing out the issue of quality from PDF figure submissions with the original manuscript. Updated figures have been revised to 1200 DPI TIFFs.

4. On lines 298-301: There are reasons beyond corresponding phenology that the nymphal tick is suspected to be the life stage most likely to spread B. burgdorferi to humans.

The authors acknowledge this point of accuracy in the discussion section. This statement has been revised (lines 443-447).

5. On line 335, an “is” is missing.

The authors thank the reviewer for pointing out this grammatical oversight. This has been revised accordingly.

Reviewer 2:

Feedback: This study examined (E)-oct-2-enal, a natural compound from donkey sebum, as a potential repellent against adult Ixodes scapularis ticks. Compared to DEET, (E)-oct-2-enal showed similar or stronger effects, reducing tick movement speed and increasing exploratory distance in a laboratory bioassay.

Comments:

1. Line 33: "Cox" should be capitalized.

The authors have revised the casing for Cox in any locations.

2. Line 72: Add an explanation for "aposematism" could aid clarity if unfamiliar to general audiences.

The authors appreciate the recommendation to make this section more accessible to a general audience. This section has been moved to the immediate beginning of the discussion section and elaborates, lines 355-357.

3. Line 112: "semiochemical" could be clarified along with its type and explain (E)-oct-2- enal falls under which type of semiochemical

The authors again appreciate the recommendation on to make this section more accessible to a general audience. This has been revised, the specific type of semiochemical (allomone) has also been included on line 98.

4. Line 152: Use "times" instead of "x" to improve readability.

The authors agree that times is more readable than x. This has been revised in all occurrences in the results section.

5. Line 292: "(P. leucopus; T. striatus)” Write full names of species when written for the first time in the manuscript

The authors have written the full species for each in line 92 at first mention.

6. Line 330: Clarify by specifying "field trial discrepancy."

The authors acknowledge that this is more appropriate phrasing. New line 475 has this revision.

7. Quality (pixels) of Figure 1 and Figure 4 should be improved.

The authors have ensured that all figures have been revised to TIFFs with 1200 DPI from original PDFs.

---

## [Decision Letter · Decision Letter 1]

8 Jan 2025

Exploiting a natural instance of vertebrate-posed chemical aposematism for tick bite prevention: Repellency of Ixodes scapularis with (E)-Oct-2-enal

PONE-D-24-39388R1

Dear Dr. Stephen M. Rich,

We’re pleased to inform you that your manuscript has been judged scientifically suitable for publication and will be formally accepted for publication once it meets all outstanding technical requirements.

Kind regards,

S Ezhil Vendan, Ph.D

Academic Editor

PLOS ONE

Additional Editor Comments (optional):

-

Reviewers' comments:

-

Reviewer's Responses to Questions

**Comments to the Author**

1. If the authors have adequately addressed your comments raised in a previous round of review and you feel that this manuscript is now acceptable for publication, you may indicate that here to bypass the “Comments to the Author” section, enter your conflict of interest statement in the “Confidential to Editor” section, and submit your "Accept" recommendation.

Reviewer #1: All comments have been addressed

Reviewer #2: All comments have been addressed

2. Is the manuscript technically sound, and do the data support the conclusions?

Reviewer #1: Yes

Reviewer #2: Yes

3. Has the statistical analysis been performed appropriately and rigorously? 

Reviewer #1: Yes

Reviewer #2: Yes

4. Have the authors made all data underlying the findings in their manuscript fully available?

Reviewer #1: Yes

Reviewer #2: Yes

5. Is the manuscript presented in an intelligible fashion and written in standard English?

Reviewer #1: Yes

Reviewer #2: Yes

6. Review Comments to the Author

Reviewer #1: The authors have addressed all of my concerns about the manuscript. The statistical analysis is much improved and provides a more clear understanding of the results. I am happy to suggest acceptance of this manuscript with no additional edits.

Reviewer #2: The author has implemented all the necessary changes, and in my opinion, the paper is now in a strong position for publication.

7. PLOS authors have the option to publish the peer review history of their article (what does this mean? ). If published, this will include your full peer review and any attached files.

**Do you want your identity to be public for this peer review?** For information about this choice, including consent withdrawal, please see our Privacy Policy .

Reviewer #1: No

Reviewer #2: No

---

## [Editor Report · Acceptance letter]

PONE-D-24-39388R1

PLOS ONE

Dear Dr. Rich,

I'm pleased to inform you that your manuscript has been deemed suitable for publication in PLOS ONE. Congratulations! Your manuscript is now being handed over to our production team.

Kind regards,

on behalf of

Dr. S Ezhil Vendan

Academic Editor

PLOS ONE